# Debiased Visual Question Answering from Feature and Sample Perspectives

**Zhiquan Wen**[1,2]**, Guanghui Xu**[1]**, Mingkui Tan**[1,3]*, **Qingyao Wu**[1]*, **Qi Wu**[4]
[1]School of Software Engineering, South China University of Technology, China
[2]PengCheng Laboratory, China
[3]Key Laboratory of Big Data and Intelligent Robot (South China University of Technology),
Ministry of Education
[4]School of Computer Science, University of Adelaide
{sewenzhiquan, sexuguanghui}@mail.scut.edu.cn,
{mingkuitan, qyw}@scut.edu.cn, qi.wu01@adelaide.edu.au

## Abstract

Visual question answering (VQA) is designed to examine the visual-textual reasoning ability of an intelligent agent. However, recent observations show that many VQA models may only capture the biases between questions and answers in a dataset rather than showing real reasoning abilities. For example, given a question, some VQA models tend to output the answer that occurs frequently in the dataset and ignore the images. To reduce this tendency, existing methods focus on weakening the language bias. Meanwhile, only a few works also consider vision bias implicitly. However, these methods introduce additional annotations or show unsatisfactory performance. Moreover, not all biases are harmful to the models. Some "biases" learnt from datasets represent natural rules of the world and can help limit the range of answers. Thus, how to filter and remove the true negative biases in language and vision modalities remain a major challenge. In this paper, we propose a method named D-VQA to alleviate the above challenges from the feature and sample perspectives. Specifically, from the feature perspective, we build a question-to-answer and vision-to-answer branch to capture the language and vision biases, respectively. Next, we apply two unimodal bias detection modules to explicitly recognise and remove the negative biases. From the sample perspective, we construct two types of negative samples to assist the training of the models, without introducing additional annotations. Extensive experiments on the VQA-CP v2 and VQA v2 datasets demonstrate the effectiveness of our D-VQA method.

## 1 Introduction

Vision-and-language tasks [5, 43, 45] require an agent to perform visual-textual reasoning, and play an important role in helping humans or intelligent robots to understand the physical world. As a typical case of the vision-and-language task, visual question answering (VQA) [8, 35, 42] aims to answer a textual question based on a given image. In this sense, VQA models should master and apply the strong reasoning ability to address the sophistical questions. However, recent studies [3, 23, 36] demonstrate that many VQA models answer the questions without reasoning. Instead, they excessively rely on superficial correlations (*i.e.,* bias) between the question and answers without considering the image. For example, some VQA models tend to answer "2" for the questions "How many . . .", since most corresponding answers in the dataset are "2". In this case, the VQA models memorise the bias but ignore the image, resulting in a poor generalisation ability.

---

*Corresponding author

35th Conference on Neural Information Processing Systems (NeurIPS 2021).

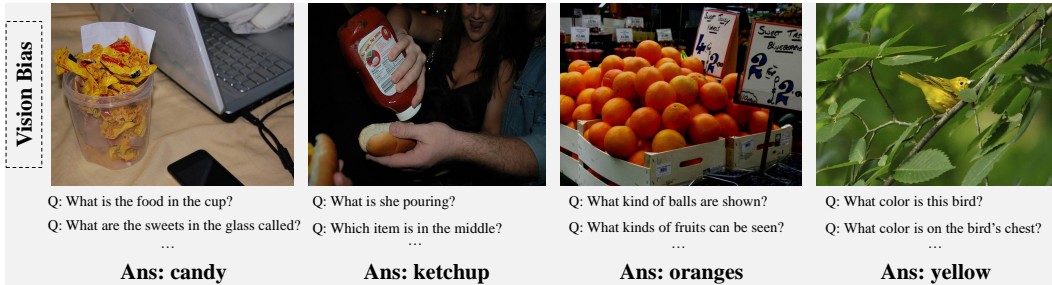

Q: What is the food in the cup?
Q: What are the sweets in the glass called?
...
**Ans: candy**

Q: What is she pouring?
Q: Which item is in the middle?
...
**Ans: ketchup**

Q: What kind of balls are shown?
Q: What kinds of fruits can be seen?
...
**Ans: oranges**

Q: What color is this bird?
Q: What color is on the bird's chest?
...
**Ans: yellow**

Figure 1: Examples of vision biases. From these examples, most of the answers to the questions usually correspond to the most salient objects/attributes in the image.

As is widely known, most machine learning datasets inevitably have biases. Thus, how to train a robust model from the biased dataset remains a major challenge. To alleviate the challenge in VQA, most existing methods mainly focus on weakening the language bias [9, 33]. Recently, Niu *et al.* [29] applied a cause-effect to alleviate language bias. However, the performance is not sufficiently encouraging, and it introduces additional parameters in the inference phase. Note that the biases also exist in the vision modality. As shown in Figure 1, many answers to the questions are usually the most salient objects/attributes in the images. In this sense, the VQA models may capture this vision bias to output an answer regarding the salient object in the image, without considering the questions. Thus, alleviating the negative effect of vision bias is also crucial. Recently, a few works [10, 17] generated counterfactual training samples by masking or transforming critical objects and words to train the models to alleviate language and vision biases. However, they require expensive annotations.

More critically, some "biases" captured from the dataset may represent the natural rule in real world, *i.e.,* commonsense knowledge. They are not harmful to the models. Instead, models benefit from them. For example, "dog" is a kind of "animal", and the colour of "oranges" is normally "orange". Thus, how to filter and remove the true negative biases in language and vision modalities is challenging.

To alleviate the above issues, we propose a method named D-VQA to overcome the negative biases in language and vision modalities from feature and sample perspectives. Specifically, **(1)** from the feature perspective, we first build a question-to-answer and a vision-to-answer branch to capture language and vision biases, respectively. Since not all biases are detrimental to the model, two unimodal bias detection modules are devised to filter and remove negative biases only. To maintain the inference efficiency, following existing methods [9, 14, 33], we do not expect to introduce additional parameters in the inference phase. To realise this expectation, we adopt a contrastive loss to approach the multimodal features to the debiased feature as close as possible. In this way, in the inference phase, we simply remove all of the additional branches, and adopt the multimodal features that have a similar representation as the debiased features to obtain accurate predictions. **(2)** From the sample perspective, the VQA model should answer the questions making full use of the information of questions and images, which can be achieved by improving the sensitivity of the model in both the images and questions. To this end, for each sample in the dataset, we construct two types of negative samples to assist the training and improve the sensitivity of the model.

Our contributions can be summarised as follows: (1) In VQA, we alleviate the biases in both language and vision modalities, from feature and sample perspectives. (2) Since not all biases are harmful, we explicitly devise the bias detection modules to filter and remove the negative biases. (3) Extensive experiments on the VQA-CP v2 and VQA v2 datasets demonstrate the effectiveness of our D-VQA.

## 2 Related Work

**Biases in Visual Question Answering (VQA).** In the vision-and-language field, biases are ubiquitous. For example, in referring expression comprehension (REF) task [15, 45, 48], Cirik *et al.* [13] argue that the benchmark datasets such as RefCOCO [24, 28] contain biases in which the referent objects are usually the most salient in the image. Thus, the models can capture the biases and obtain high accuracy without the queries. Inevitably, the issues of bias also exist in VQA. As demonstrated by [2, 3, 21, 36], existing VQA benchmark datasets such as the VQA v2 [18] have biases, namely, a

shortcut exists between the questions and answers. For example, one simply answering "tennis" for the question "What sports . . . " will obtain high accuracy, because the model is trained and tested on the COCO image dataset that has many "tennis" related images. Moreover, most methods consider language bias only, while ignoring the vision bias. As discussed in [29], the questions usually involve the objects seen in the image. In addition, as shown in Figure 1, the most salient objects/attributes are usually chosen to be the answers. In this sense, the VQA models may also capture this vision bias. Thus, in this paper, we consider alleviating the biases in language and vision modalities.

**Overcoming biases in VQA.** Recently, many methods [9, 10, 16, 17, 22, 26, 27, 29, 33, 36, 39, 40, 44, 49] have been proposed to overcome the biases in VQA. Formally, these methods can be categorised into two classes, namely, with and without data augmentations. Specifically, non-augmentation-based methods [9, 16, 22, 26, 29, 33, 36, 44] seek to reduce the language biases explicitly or improve attention on the image, while augmentation-based methods [1, 10, 17, 27, 38, 40, 49] seek to balance the biased dataset for unbiased training.

For the non-augmentation-based methods, Ramakrishnan *et al.* [33] adopt adversarial learning between the base VQA model and the question-only model to prevent the VQA model from capturing language bias as much as possible. Inspired by [33], Cadene *et al.* [9] dynamically adjust the weight of the samples based on how biased the samples are. Moreover, some methods introduce human based visual [36] and text [44] explanations to strengthen the visual grounding. However, these methods require human annotations that are hard to obtain. Apart from the above, Niu *et al.* [29] introduce the cause-effect to look at language bias and propose a counterfactual inference framework to reduce the biases, which unfortunately introduces additional parameters in inference.

For the augmentation-based methods, to promote the VQA model to focus on the critical objects and words, Chen *et al.* [10] propose a CSS method to produce massive counterfactual samples by masking the critical objects and words, and assign corresponding ground-truth answers. To make full use of the samples, Liang *et al.* [27] model the relationships among original, factual and counterfactual samples to promote learning high-level features. Moreover, the Mutant [17] generates the samples by semantic transformations of the original images or questions. Without introducing additional annotations, some methods [38, 49] build negative samples from the samples available to balance the dataset. In this paper, we overcome the bias from sample and feature perspectives. Specifically, from the sample perspective, inspired by [38, 49], we construct two types of negative samples to assist the training of the model. From the feature perspective, we build question-to-answer and vision-to-answer branches to capture the language and vision biases, respectively. Next, we devise two unimodal bias detection modules to recognise and remove the negative biases.

**Contrastive Learning** [41] aims to learn the high-level representation by maximising the mutual information between the input samples and positive samples, which has been applied in many fields [27, 32, 47]. Specifically, some contrastive learning methods [11, 19] adopt contrastive learning to train the models by decreasing the distance between the feature representations of different augmented views of the same images, while increasing the distance between different images. Liang *et al.* [27] adopt contrastive learning to approach the feature representations of the original samples to the factual samples, while keeping away from that between the original and counterfactual samples. In this paper, we adopt contrastive learning to push the multimodal feature to approach the debiased features, and meanwhile to far away from the negative biased features.

## 3    Proposed Method

As demonstrated by [3, 23, 36], many VQA models address the questions by exploiting the superficial correlation (*i.e.,* bias) between the questions and answers without considering the images, resulting in poor generalisation ability. (*i.e.,* performing worse on an out-of-distribution (OOD) dataset, *e.g.,* VQA-CP [3]). Moreover, bias also exists between the images and answers (as shown in Figure 1), which is ignored by most previous methods [9, 14, 33, 38, 49]. However, some "biases" learnt from the dataset indicate the natural rule of the world, such as commonsense knowledge. Thus, how to filter and remove the negative biases in language and vision modalities is challenging.

In this paper, we alleviate the negative effect of both language and vision biases from feature and sample perspectives. An overview of our D-VQA method is shown in Figure 2. Specifically, from the feature perspective (see Section 3.2), we seek to obtain multimodal features that reduce the negative biases as much as possible. To this end, we construct a question-to-answer and vision-to-answer

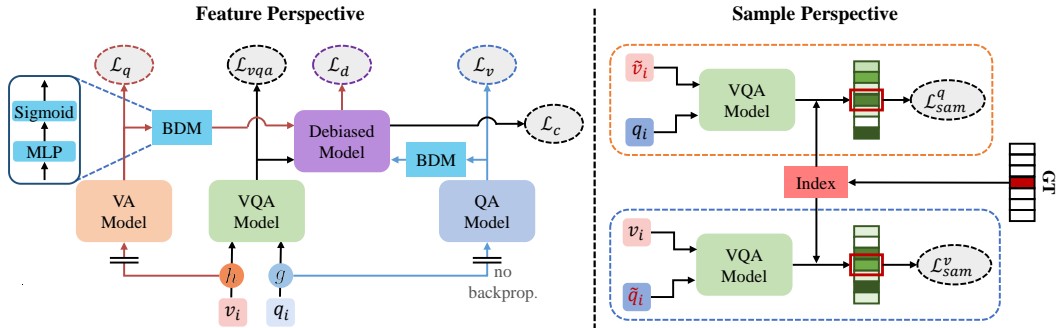

Figure 2: Overview of our D-VQA method. Our method alleviates the biases from feature and sample perspectives. From the feature perspective, we introduce question-to-answer, vision-to-answer, and debiased branches to help reduce the negative biases. BDM is the bias detection module. Note that $\mathcal{L}_d$ does not optimise the VQA model. From the sample perspective, we construct two types of negative samples to assist the model's training. $(v_i, q_i)$ denotes a positive sample, while $(\tilde{v}_i, q_i)$ and $(v_i, \tilde{q}_i)$ are negative samples. GT is the ground-truth answer of $(v_i, q_i)$. $h$ and $g$ are the feature extractors.

branch to capture the unimodal biases. Moreover, we devise two unimodal bias detection modules to explicitly recognise and remove the true negative biases. That is, we obtain debiased features by simply removing negative biased features from multimodal features. However, using the debiased feature in inference will introduce additional parameters. To address this issue, inspired by contrastive learning [11, 19], we adopt contrastive loss to approach the multimodal features to the debiased features while keeping away from the negative biased features. In this way, in the inference phase, we discard all of the additional branches and adopt the multimodal features to obtain accurate predictions.

From the sample perspective (see Section 3.3), the VQA models are supposed to simultaneously focus on language and vision modalities when answering the questions. Specifically, given a question-image pair (positive sample), we construct two types of negative samples by randomly sampling an image or question in mini-batch data. The predicted possibility for the ground-truth answer should be minimised if negative samples are used. In this way, VQA models are able to capture the two modalities information to generate the answer instead of focusing on one modality only. Based on the above discussion, our D-VQA method is able to alleviate the negative effect of the biases in language and vision modalities and improve the model performance on the OOD dataset (*e.g.,* VQA-CP v2 [3]).

## 3.1 Preliminaries

Visual question answering (VQA) seeks to answer the textual questions based on the given images. Following traditional VQA methods [4, 7, 8, 25, 46], we regard the VQA task as a multiclass classification problem. Specifically, given a dataset $\mathcal{D} = \{(v_i, q_i, a_i)\}_{i=1}^{N}$ with $N$ triplets, where the $i$-th image $v_i \in \mathcal{V}$ (image set), question $q_i \in \mathcal{Q}$ (question set) and answers $a_i \in \mathcal{A}$ (answer set), one seeks to train a model to learn a mapping function $\mathcal{F} : v_i \times q_i \to \mathbb{R}^{|\mathcal{A}|}$ to generate an accurate answer distribution over the answer set $\mathcal{A}$. To be specific, $\mathcal{F}$ usually contains three parts: (1) Feature extraction. It requires building two feature extractors $h$ and $g$ to extract the features of the images and questions, respectively. For one triplet in $\mathcal{D}$, the image extractor $h : v_i \to \mathbb{R}^{n_v \times d_v}$ outputs $n_v$ vectors with a dimension $d_v$, and the question extractor $g : q_i \to \mathbb{R}^{n_q \times d_q}$ outputs $n_q$ vectors with a dimension $d_q$. (2) Multimodal feature fusion. Since the image and question features come from different modalities, it requires designing a feature fusion function $f$ to fuse both of the image and question features, where $f : \mathbb{R}^{n_v \times d_v} \times \mathbb{R}^{n_q \times d_q} \to \mathbb{R}^{d_h}$. (3) Answer prediction. It devises a classifier $c : \mathbb{R}^{d_h} \to \mathbb{R}^{|\mathcal{A}|}$ for producing the answer prediction. Note that most of the classic VQA models [4, 7, 8, 25, 46] follow this paradigm, which can be formulated as:

$$\mathcal{F}(\mathcal{A}|v_i, q_i) = c(\mathbf{f}), \quad \text{where} \ \mathbf{f} = f(h(v_i), g(q_i)). \tag{1}$$

Formally, the parameters of the VQA model can be optimised by a binary cross-entropy loss:

$$\mathcal{L}_{vqa} = -\frac{1}{N} \sum_{i=1}^{N} \mathbf{y}_i \log(\sigma(\mathcal{F}(\mathcal{A}|v_i, q_i))) + (1 - \mathbf{y}_i)\log(1 - \sigma(\mathcal{F}(\mathcal{A}|v_i, q_i))), \tag{2}$$

where $\sigma(\cdot)$ is the sigmoid activation function, and $\mathbf{y}_i$ is the target score of each answer for the $i$-th sample, denoted as $\mathbf{y}_i = min(\frac{\#votes}{3}, 1)$, where $\#votes$ is the number of each answer that humans annotated for the question $q_i$. However, Agrawal *et al.* [3] show that many VQA models tend to capture unimodal biases (*e.g.,* the shortcut between the questions and answers) from the dataset. To alleviate this issue, we consider reducing the negative biases from **feature and sample perspectives**.

## 3.2 Reducing biases from the feature perspective

In this part, we seek to reduce the negative effect of the biases from the feature perspective, namely, decrease the negative biases from the multimodal features $\mathbf{f}$. Before that, we first need to capture the biases in both language and vision modalities.

**Unimodal model to capture the biases.** To capture the biases in a VQA dataset, one intuitive way is to train a unimodal model as a branch that takes **only one** modal feature as input. Specifically, we construct a question-to-answer and vision-to-answer branch to capture language and vision biases, respectively (we conduct the ablation studies on each branch in Table 2). Since the structures of the two branches are similar, we describe the question-to-answer branch only and obtain the vision-to-answer branch by simply replacing the subscript of $q$ with $v$.

Specifically, our question-to-answer model $\mathcal{F}_q : q_i \to \mathbb{R}^{|\mathcal{A}|}$ contains a multi-layer perceptron (MLP) $m_q : \mathbb{R}^{n_q \times d_q} \to \mathbb{R}^{d_h}$ and a classifier $c_q : \mathbb{R}^{d_h} \to \mathbb{R}^{|\mathcal{A}|}$. Based on the above, the question-to-answer branch can be formulated as:

$$\mathcal{F}_q(\mathcal{A}|q_i) = c_q(\mathbf{f}_q), \quad \text{where} \quad \mathbf{f}_q = m_q(g(q_i)). \tag{3}$$

By training the question-to-answer model using binary cross-entropy loss $\mathcal{L}_q$ on a VQA dataset, the model is able to capture the biases between the questions and answers in the dataset. Thus, the feature $\mathbf{f}_q$ contains language biases. We can obtain the vision biased feature $\mathbf{f}_v$ in a similar way.

**Recognise and obtain the negative biases.** We argue that not all biases in the dataset are harmful to the VQA models. Some "biases" captured from the dataset contain commonsense knowledge that may be helpful for the VQA models. Thus, we devise a simple but effective bias detection module (we evaluate the bias detection modules in Table 3) to detect the negative biases that need to be removed in the feature $\mathbf{f}_q$. Formally, our bias detection module is constructed by an MLP $m_b : \mathbb{R}^{d_h} \to \mathbb{R}$ and a sigmoid activation function $\sigma(\cdot)$, which can be formulated as:

$$b_q = \sigma(m_b(\mathbf{f}_q)). \tag{4}$$

Based on the weight $b_q$, the negative language biased features can be represented as $\mathbf{f}_{bias}^q = b_q \cdot \mathbf{f}_q$. The negative vision biased feature $\mathbf{f}_{bias}^v$ is obtained similarly. To obtain the global negative biased feature $\mathbf{f}_{bias}$, we adopt an attention mechanism to fuse two negative unimodal biased features:

$$\mathbf{a} = \text{softmax}(m_a([\mathbf{f}_{bias}^q; \mathbf{f}_{bias}^v])), \quad \mathbf{f}_{bias} = \mathbf{a}^\top [\mathbf{f}_{bias}^q; \mathbf{f}_{bias}^v], \tag{5}$$

where $m_a : \mathbb{R}^{2 \times d_h} \to \mathbb{R}^2$ is an MLP, and $[\cdot ; \cdot]$ denotes for stacking.

**Obtain debiased features.** Intuitively, the debiased features can be obtained by removing negative biased features $\mathbf{f}_{bias}$ from the multimodal features $\mathbf{f}$, which can be represented as $\mathbf{f}_d = \mathbf{f} - \mathbf{f}_{bias}$. With the feature $\mathbf{f}_d$ as input, we also introduce another debiased branch that contains an MLP $m_d : \mathbb{R}^{d_h} \to \mathbb{R}^{d_h}$ and an independent classifier $c_d : \mathbb{R}^{d_h} \to \mathbb{R}^{|\mathcal{A}|}$, which can be formulated as:

$$\mathcal{F}_{debias}(\mathcal{A}|v_i, q_i) = c_d(\mathbf{f}_c), \quad \text{where} \quad \mathbf{f}_c = m_d(\mathbf{f}_d). \tag{6}$$

The parameters of the debiased branch are optimised by the binary cross-entropy loss $\mathcal{L}_d$. However, if we adopt the debiased branch as the target model, it will introduce additional parameters in the inference phase. To address this issue, an intuitive idea is to make the multimodal features $\mathbf{f}$ and debiased features $\mathbf{f}_c$ generated from the debiased branch as similar as possible. To accomplish this task, inspired by the contrastive learning [11, 19], we consider forcing the multimodal features $\mathbf{f}$ to near the debiased features $\mathbf{f}_c$ and away from the negative biased features $\mathbf{f}_{bias}$.

To achieve the abovementioned aim, our contrastive loss can be formulated as follows:

$$\mathcal{L}_c = -\frac{1}{N} \sum_{i=1}^{N} \log(\frac{e^{s(\mathbf{f}, \mathbf{f}_c)}}{e^{s(\mathbf{f}, \mathbf{f}_c)} + e^{s(\mathbf{f}, \mathbf{f}_{bias})}}), \quad \text{where} \quad s(\mathbf{f}_1, \mathbf{f}_2) = \frac{\mathbf{f}_1^\top \mathbf{f}_2}{\|\mathbf{f}_1\| \cdot \|\mathbf{f}_2\|}. \tag{7}$$

$s(\cdot, \cdot)$ denotes a scoring function, of which the higher the value is, the higher the similarity between two features. In practice, we adopt the cosine similarity as the scoring function. By minimising the loss $\mathcal{L}_c$, the multimodal features $\mathbf{f}$ are able to approach the debiased features $\mathbf{f}_c$ while avoiding the negative biased features $\mathbf{f}_{bias}$. In this way, the multimodal features $\mathbf{f}$ have a similar representation with the debiased features $\mathbf{f}_c$, to some extent. Note that all of the additional branches participate in the training phase only. In the inference phase, we remove all of the additional branches and adopt the base VQA model to obtain accurate predictions.

### 3.3 Reducing biases from the sample perspective

Since the biases usually embody in capturing the shortcut between one modal information and the answer, it is necessary to improve the attention of the VQA models on language and vision modalities when answering the questions. That is, increasing the sensitivity of the VQA models on the information of both modalities. To achieve this, we consider constructing two types of negative samples for each sample in the dataset to assist the training of the model. Specifically, inspired by [38, 49], for each positive sample $(v_i, q_i, a_i)$, we construct the negative samples by randomly sampling question $\tilde{q}_i$ and image $\tilde{v}_i$ in mini-batch data as $(\tilde{v}_i, q_i, a_i)$ and $(v_i, \tilde{q}_i, a_i)$. Note that $\tilde{v}_i$ and $\tilde{q}_i$ are sampled separately. The ablation studies regarding the negative samples are in Table 2.

Intuitively, in VQA, one can answer the question only if the question and image are corresponding. In contrast, when taking the negative samples as input, the VQA model cannot answer the question correctly, which is able to be achieved by minimising the possibility of predicting the ground truth answer of the positive sample. Since the training procedure of these two types of negative samples is the same, we only describe the training details of $(\tilde{v}_i, q_i, a_i)$. Formally, the loss can be defined as:

$$\mathcal{L}_{sam}^q = \text{softmax}(\mathcal{F}(\mathcal{A}|\tilde{v}_i, q_i))[k], \tag{8}$$

where $k$ is the index of the answer $a_i$ in the answer set $\mathcal{A}$. Minimising $\mathcal{L}_{sam}^q$ encourages the prediction of the VQA model to keep away from the ground truth answer of the positive sample, because of the lack of supporting visual information. In this way, this type of negative samples is able to promote the VQA model to focus on the images when answering the question, which improves the sensitivity of the model on the images. Similarly, another type of negative samples $(v_i, \tilde{q}_i, a_i)$ is able to improve the sensitivity of the model on the questions. The loss of the sample perspective can be defined as $\mathcal{L}_{sam} = \lambda_q \mathcal{L}_{sam}^q + \lambda_v \mathcal{L}_{sam}^v$ (the ablation studies on $\lambda_q$ and $\lambda_v$ are in Table 5). Note that we do not introduce additional annotations, instead, we make full use of the training samples available.

### 3.4 Learning D-VQA with overall loss

In total, our method contains three types of losses, namely, traditional VQA loss (*i.e.,* binary cross-entropy loss), contrastive loss and sample perspective loss, which can be formulated as:

$$\mathcal{L} = \mathcal{L}_{vqa} + L_q + \mathcal{L}_v + \mathcal{L}_d + \mathcal{L}_c + \mathcal{L}_{sam}, \tag{9}$$

where $\mathcal{L}_{\{vqa,q,v,d\}}$ are the binary cross-entropy loss based on the same ground-truth answer, corresponding to the {base, question-to-answer, vision-to-answer, debiased} branches, respectively.

## 4 Experiments

### 4.1 Datasets and compared methods

**Datasets.** To demonstrate the effectiveness of our D-VQA method, we evaluate it on the out-of-distribution benchmark dataset VQA-CP (Visual Question Answering under Changing Priors) v2 [3] and IID dataset VQA v2 [18] validation set based on the standard evaluation metric [6]. Since the VQA v2 [18] dataset has strong biases, to evaluate the generalisation ability of the VQA model, Agrawal *et al.* [3] created the out-of-distribution (OOD) dataset named VQA-CP v2. Specifically, they constructed VQA-CP v2 by re-organising the training and validation sets of the VQA v2 dataset, where the answer distributions in training and test sets of VQA-CP v2 are different. The training set of VQA-CP v2 contains approximately 121k images and 245k questions, while the test set contains approximately 98k images and 220k questions.

**Compared methods.** We regard both augmentation-based and non-augmentation-based methods as compared methods. Specifically, (1) for the non-augmentation-based methods, we compare our

D-VQA method with AdvReg [33], RUBI [9], LMH [14], DLR [22], VGQE [26], HINT [36], SCR [44], RMFE [16] and CF-VQA [29]. Moreover, (2) augmentation-based methods such as CVL [1], Unshuffling [40], CSS [10], RandImg [38], SSL-VQA [49], CSS+CL [27], and Mutant [17] are considered compared methods. Note that some augmentation-based methods such as CSS and Mutant construct the counterfactual samples using additional annotations, while our D-VQA does not introduce additional annotations.

## 4.2 Implementation details

Our method is model-agnostic and can be applied to different backbones of VQA models. To better demonstrate the effectiveness of our D-VQA, we conduct experiments based on different backbones, including UpDn [4], SAN [46] and LXMERT [37]. Following previous works [9, 33, 49], we adopt Faster-RCNN [34] that are pretrained by [4] to extract the object features. Specifically, we extract the top-36 object features for each image and the dimension of each object feature is 2048. Moreover, all the questions are processed into the same length (*i.e.,* 14), and then each word in the question is encoded by GloVe [31] embedding with a dimension of 300. A single layer GRU [12] is used to obtain the feature of the question with a dimension of 1280.

Inspired by SSL-VQA [49], we add one Batch Normalisation [20] layer before the classifier of UpDn [4], and then train our D-VQA for 25 epochs. Specifically, we train all the branches with the binary cross-entropy loss and contrastive loss over the training process, and the sample perspective loss is introduced at the 13-th epoch. We adopt Adam optimiser with the initial learning rate of $1e$-3, and the learning rate decreases by half every 5 epochs after 10 epochs. The batch size is set to 256.

For the backbone of LXMERT [37], we load the pre-trained LXMERT model from the official GitHub repository[2], and then perform the downstream VQA-CP task with our D-VQA method. Specifically, we train LXMERT + D-VQA for 10 epochs, and the sample perspective loss is introduced at the 7-th epoch. The batch size is 32, and the learning rate is $1e$-5. We implement our entire method based on PyTorch [30], and the model is trained with one Titan Xp GPU. The source code and the pre-trained models are available at https://github.com/Zhiquan-Wen/D-VQA.

## 4.3 Evaluation on the VQA-CP v2 and VQA v2 datasets

**Quantitative results.** We compare our D-VQA method with the state-of-the-art methods on the VQA-CP v2 test set and VQA v2 validation set, and report the experimental results in Table 1. From these results, we have the following observations. On the VQA-CP v2 dataset, (1) augmentation-based methods surpass non-augmentation-based methods by a large margin (57.59% *vs.* 54.55% and 61.72% *vs.* 54.55%). These results demonstrate that reducing bias by focusing on data is more effective than other non-augmentation-based methods. (2) Our D-VQA outperforms all the compared methods, either augmentation-based or non-augmentation-based. Specifically, our D-VQA surpasses CF-VQA [29], SSL-VQA [49] and Mutant [17] by approximately 8%, 4%, and 0.2%, respectively. Note that some augmentation-based methods such as Mutant [17], CSS [10] and CSS+CL [27] generate counterfactual training samples by masking or transforming critical words and objects from the samples in the dataset, which introduces additional annotations. In our method, without using additional annotations, we construct the negative samples based on the available samples, which fully makes use of the samples in the dataset. Nevertheless, our D-VQA still performs better than Mutant, which greatly demonstrates the effectiveness of our D-VQA. (3) For the question type of "Num", our D-VQA outperforms the compared methods by a large margin, and achieves comparable performance on the remaining question types. These results further verify the effectiveness of our D-VQA.

On the validation set of the VQA v2 dataset, most methods perform worse than UpDn [4], while our D-VQA surpasses all compared methods. These results show that our D-VQA performs better in both IID (VQA v2) and OOD (VQA-CP v2) datasets, indicating the superiority of our D-VQA.

**Qualitative results.** To further demonstrate the effectiveness of our method on the OOD dataset, we provide qualitative results on the test set of the VQA-CP v2 dataset in Figure 3. From these results, UpDn and SSL-VQA cannot find the target objects accurately, and thus output the wrong answers or have low scores in the ground-truth answer. Our D-VQA method locates the target objects with high attention weight, and then makes an accurate prediction.

---

[2]https://github.com/airsplay/lxmert

Table 1: Comparison with the state-of-the-art methods on the VQA-CP v2 test set and VQA v2 validation set in terms of Accuracy (%). The backbone of all models is UpDn [4]. Overall best scores are **bold**, and the second best of overall scores of ours are underlined. **I – IV** denote non-augmentation-based methods using human annotations, non-augmentation-based methods without using human annotations, augmentation-based methods adopting additional annotations, and augmentation-based methods without introducing additional annotations, respectively.

| Case | Model | VQA-CP v2 test (%) | | | | VQA v2 val (%) | | | |
|------|-------|------|--------|------|-------|------|--------|------|-------|
| | | All | Yes/No | Num | Other | All | Yes/No | Num | Other |
| – | SAN [46] | 24.96 | 38.35 | 11.14 | 21.74 | 52.41 | 70.06 | 39.28 | 47.84 |
| | GVQA [3] | 31.30 | 57.99 | 13.68 | 22.14 | 48.24 | 72.03 | 31.17 | 34.65 |
| | UpDn [4] | 39.74 | 42.27 | 11.93 | 46.05 | 63.48 | 81.18 | 42.14 | 55.66 |
| **I** | AttAlign [36] | 39.37 | 43.02 | 11.89 | 45.00 | 63.24 | 80.99 | 42.55 | 55.22 |
| | HINT [36] | 46.73 | 67.27 | 10.61 | 45.88 | 63.38 | 81.18 | 42.99 | 55.56 |
| | SCR [44] | 48.47 | 70.41 | 10.42 | 47.29 | 62.30 | 77.40 | 40.90 | 56.50 |
| **II** | AdvReg [33] | 41.17 | 65.49 | 15.48 | 35.48 | 62.75 | 79.84 | 42.35 | 55.16 |
| | RUBi [9] | 44.23 | 67.05 | 17.48 | 39.61 | - | - | - | - |
| | DLR [22] | 48.87 | 70.99 | 18.72 | 45.57 | 57.96 | 76.82 | 39.33 | 48.54 |
| | VGQE [26] | 48.75 | - | - | - | 64.04 | - | - | - |
| | LMH [14] | 52.01 | 72.58 | 31.12 | 46.97 | 56.35 | 65.06 | 37.63 | 54.69 |
| | RMFE [16] | 54.55 | 74.03 | 49.16 | 45.82 | - | - | - | - |
| | CF-VQA [29] | 53.55 | **91.15** | 13.03 | 44.97 | 63.54 | **82.51** | 43.96 | 54.30 |
| **III** | CSS [10] | 58.95 | 84.37 | 49.42 | 48.21 | 59.91 | 73.25 | 39.77 | 55.11 |
| | CSS+CL [27] | 59.18 | 86.99 | 49.89 | 47.16 | 57.29 | 67.27 | 38.40 | 54.71 |
| | Mutant [17] | 61.72 | 88.90 | 49.68 | **50.78** | 62.56 | 82.07 | 42.52 | 53.28 |
| **IV** | CVL [1] | 42.12 | 45.72 | 12.45 | 48.34 | - | - | - | - |
| | Unshuffling [40] | 42.39 | 47.72 | 14.43 | 47.24 | 61.08 | 78.32 | 42.16 | 52.71 |
| | RandImg [38] | 55.37 | 83.89 | 41.60 | 44.20 | 57.24 | 76.53 | 33.87 | 48.57 |
| | SSL-VQA [49] | 57.59 | 86.53 | 29.87 | 50.03 | 63.73 | - | - | - |
| | **D-VQA** | **61.91** | 88.93 | **52.32** | 50.39 | **64.96** | 82.18 | **44.05** | **57.54** |

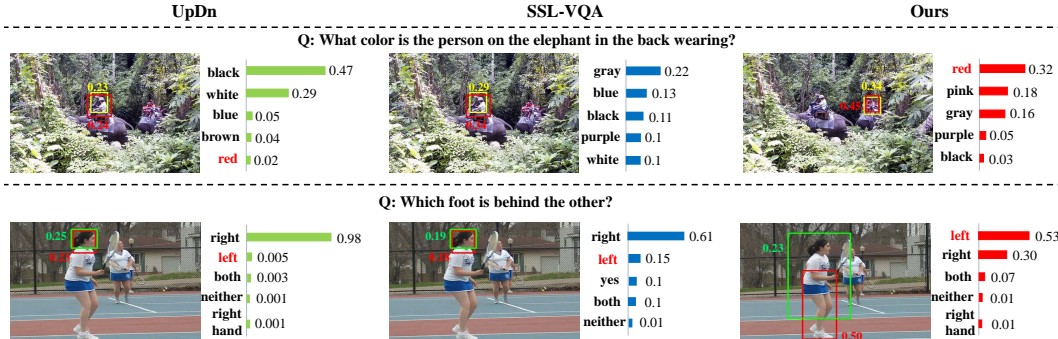

Figure 3: Qualitative comparison among UpDn [4], SSL-VQA [49], and our D-VQA on the VQA-CP v2 test set. For each example, we put the bounding box with the top-2 attention weights in the image and show the answers with the top-5 predictions. The bold, red answer is the ground-truth answer.

## 4.4 Ablation studies of D-VQA

**Effect of each component on the model performance.** To demonstrate the effectiveness of each component in our D-VQA, we conduct ablation studies on VQA-CP v2 regarding the components of D-VQA, and show the experimental results in Table 2. From these results, we have the following observations: (1) The question-to-answer and vision-to-answer branches promote the model performance (refer to Lines 4-6 of the Table 2), which demonstrates that these two branches are able to capture the corresponding negative biases to be removed. (2) Both types of negative samples

Table 2: Effect of each component of our D-VQA on the model performance. We show the results on the VQA-CP v2 dataset in terms of Accuracy (%). We use UpDn[4] as the backbone model, where † denotes our re-implementation of the baseline.

| | $(\tilde{v}_i, q_i)$ | $(v_i, \tilde{q}_i)$ | question-to-answer | vision-to-answer | VQA-CP v2 test (%) |
|---|---|---|---|---|---|
| 1 | | | | | 41.53 |
| 2 | √ | | | | 57.59 |
| 3 | | √ | | | 41.60 |
| 4 | √ | √ | | | 60.24 |
| 5 | √ | √ | √ | | 61.18 |
| 6 | √ | √ | √ | √ | **61.91** |

Table 3: Effect of the bias detection modules on the VQA-CP v2 dataset. "base" denotes D-VQA without the bias detection modules, while "$\text{det}_{\{v,q\}}$" denotes the vision/language bias detection module.

| | Model | VQA-CP v2 test (%) |
|---|---|---|
| 1 | base | 59.93 |
| 2 | base + $\text{det}_v$ | 59.98 |
| 3 | base + $\text{det}_q$ | 60.28 |
| 4 | D-VQA | **61.91** |

Table 4: Effect of different backbones on the model performance. We report the experimental results on the VQA-CP v2 dataset in terms of Accuracy (%). † denotes the re-implementation of the baseline.

| Model | Yes/No | Number | Other | Overall | Gap$\Delta \uparrow$ |
|---|---|---|---|---|---|
| SAN† [46] | 38.44 | 12.91 | **46.65** | 39.11 | |
| **SAN + D-VQA** | **84.12** | **26.12** | 46.56 | **54.39** | **+15.28** |
| UpDn† [4] | 43.45 | 13.64 | 48.18 | 41.53 | |
| **UpDn + D-VQA** | **88.93** | **52.32** | **50.39** | **61.91** | **+20.38** |
| LXMERT [37] | 42.84 | 18.91 | 55.51 | 46.23 | |
| **LXMERT + D-VQA** | **80.43** | **58.57** | **67.23** | **69.75** | **+23.52** |

are able to improve the model performance, and only $(\tilde{v}_i, q_i, a_i)$ improves substantially, while only $(v_i, \tilde{q}_i, a_i)$ is marginally improved. Additionally, combining these two types of negative samples further promotes the model performance (refer to Lines 1-4 of the Table 2). From these results, we surmise that language bias is dominant compared with vision bias, while vision bias cannot be ignored. In total, these results demonstrate the effectiveness of each component in our D-VQA method on the model performance.

**Evaluation of the bias detection modules.** The bias detection modules are devised to determine how many negative biases should be removed. To evaluate the effect of the bias detection modules (language and vision bias detection modules), we conduct experiments on the VQA-CP v2 dataset about removing the bias detection modules, and the experimental results are reported in Table 3. We have the following observations from these results: (1) both of the language and vision bias detection modules promote the model performance, while the language bias detection module has a more positive effect than the vision bias detection module (refer to Lines 1-3 of Table 3). These results demonstrate the necessity to recognise the negative bias, and then remove it. (2) Combining language and vision bias detection modules is able to further promote the model performance by a large margin (refer to Lines 2-4 of Table 3), which demonstrates the necessity and effectiveness of recognising and removing the negative biases in both language and vision modalities.

**Effect of different backbones.** To demonstrate the effectiveness of our D-VQA on different backbones, we conduct experiments on the VQA-CP v2 dataset using different backbones (*i.e.,* SAN [46], UpDn [4], and LXMERT [37]), and report the experimental results in Table 4. From these results, we find that D-VQA is able to improve the model performance substantially regardless of the backbone, which demonstrates that our method is model-agnostic, embodying the superiority of our D-VQA.

**Evaluation of different combinations between $\lambda_q$ and $\lambda_v$.** From the experimental results in Lines 2-3 of Table 2, $(\tilde{v}_i, q_i)$ has a more positive effect than $(v_i, \tilde{q}_i)$. In this sense, $\lambda_q$ should have a higher weight than $\lambda_v$. To further make a trade-off between $\mathcal{L}_{sam}^q$ and $\mathcal{L}_{sam}^v$, we conduct experiments on VQA-CP v2 regarding different combinations of $\lambda_q$ and $\lambda_v$, and the experimental results are reported in Table 5. From these results, the highest performance is obtained in our D-VQA when $\lambda_q$ *vs.* $\lambda_v$ is 1.0 : 0.7, and the performance declines for any $\lambda_v$ higher or lower than 0.7, which demonstrates a suitable ratio between $\lambda_q$ and $\lambda_v$ will obtain better performance in our method.

Table 5: Effect of different combinations between $\lambda_q$ and $\lambda_v$. We report the experimental results in terms of Accuracy (%).

| Model | $\lambda_q$ *vs.* $\lambda_v$ | VQA-CP v2 test (%) |
|---|---|---|
| | 1 : 0.1 | 60.74 |
| | 1 : 0.3 | 61.74 |
| D-VQA | 1 : 0.5 | 61.40 |
| | 1 : 0.7 | **61.91** |
| | 1 : 1.0 | 61.63 |

Table 6: Effect of different scales of the training data of VQA-CP v2 on the model performance. We report the results in terms of Accuracy (%).

| Model | Proportion of Training Set | | | | |
|---|---|---|---|---|---|
| | 20% | 40% | 60% | 80% | 100% |
| UpDn$^\dagger$ [4] | 36.22 | 38.90 | 39.40 | 40.61 | 41.53 |
| SSL-VQA [49] | 52.71 | 54.42 | 56.83 | 57.31 | 57.59 |
| **D-VQA** | **52.94** | **56.74** | **58.31** | **59.05** | **61.91** |

**Performance on different scales of the dataset.** To further demonstrate the superiority of our method, we conduct experiments regarding different scales of the training data on VQA-CP v2. Specifically, the model is trained based on the randomly sampled data (*i.e.,* from 20% to 80% of the original training data) coming from the original training data, and then is evaluated on the standard test set. The experimental results are reported in Table 6. From these results, we find that (1) our D-VQA achieves the best results compared with the state-of-the-art method SSL-VQA [49] and the baseline method UpDn [4] regardless of the scales of the training data. (2) The performance gap between our D-VQA and compared methods gradually increases with the number of training samples. These results demonstrate the effectiveness of our method on different scales of the training data.

## 5 Conclusion

In this paper, we have proposed a novel method named D-VQA to overcome the negative biases in both language and vision modalities. In our D-VQA, we alleviate the negative effect of the biases from the feature and sample perspectives. Specifically, from the feature perspective, we introduce a question-to-answer and a vision-to-answer branch to capture the corresponding unimodal biases. Moreover, not all biases are harmful to the VQA models. Some "biases" in the dataset indicate the natural rule of the world, *e.g.,* commonsense knowledge. Thus, we explicitly devise the bias detection modules to detect and remove the negative biases. From the sample perspective, we construct two types of negative samples to assist the training and improve the sensitivity of the model on both language and vision modalities. Extensive experiments on the VQA-CP v2 dataset show that our D-VQA achieves a new state-of-the-art result, which demonstrates its effectiveness. Note that our method is model-agnostic (refer to Table 4), and can be embedded into other backbones to overcome the negative biases. The current work focuses on the VQA task only, we will extend our approach to other multimodal tasks in future works, *e.g.,* referring expression comprehension. Moreover, although our D-VQA achieves state-of-the-art performance on most types of questions (see Table 1 of this paper), there is much room for improvement on questions such as "Num" and "Other".

## 6 Acknowledgement

We would like to thank all the anonymous reviewers for their constructive comments and suggestions. This work was partially supported by Ministry of Science and Technology Foundation Project 2020AAA0106900, National Natural Science Foundation of China (NSFC) 61876208, Key-Area Research and Development Program of Guangdong Province 2018B010108002, Key Realm R&D Program of Guangzhou 202007030007, Program for Guangdong Introducing Innovative and Enterpreneurial Teams 2017ZT07X183.

## 7 Broader Impact

This paper introduces a novel D-VQA method to alleviate the biases in Visual Question Answering (VQA). Overcoming the bias can promote the VQA model to use real reasoning ability to address sophisticated questions. Therefore, this research can promote the development of the robot *e.g.,* dialogue robot, and facilitate people's daily lives. The failure of the debiased technique may result in the collapse of the VQA system in environments that have never been seen. Moreover, we evaluate our method on benchmark out-of-distribution dataset (OOD) and demonstrate better debiased ability.

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
