# Debiased Visual Question Answering from Feature and Sample Perspectives

**Zhiquan Wen**[1,2], **Guanghui Xu**[1], **Mingkui Tan**[1,3*], **Qingyao Wu**[1*], **Qi Wu**[4]
[1]School of Software Engineering, South China University of Technology, China
[2]PengCheng Laboratory, China
[3]Key Laboratory of Big Data and Intelligent Robot (South China University of Technology),
Ministry of Education
[4]School of Computer Science, University of Adelaide
{sewenzhiquan, sexuguanghui}@mail.scut.edu.cn,
{mingkuitan, qyw}@scut.edu.cn, qi.wu01@adelaide.edu.au

In the supplementary materials, we provide the training method and more experimental results of our D-VQA. We organise the supplementary materials as follows:

- In Section A, we provide the training method of our D-VQA.
- In Section B, we provide more ablation studies of our D-VQA.
- In Section C, we provide more visualisation results of our D-VQA on VQA-CP v2 [1].

## A  Training method

To describe our D-VQA clearly, we provide the training method in Algorithm 1. Specifically, we first construct a question-to-answer, a vision-to-answer, and a debiased branch. We then forward all of the branches with the positive samples to calculate the feature perspective losses, namely $L_{\{vqa,q,v,d,c\}}$. When the training epoch is higher than threshold $\tau$ that indicates when to introduce the sample perspective loss, we construct the negative samples without introducing additional annotations, and then forward the base model with the negative samples and obtain the sample perspective loss $L_{sam}$. Finally, we update all of the branches based on the overall loss $L$.

## B  More Ablation studies

To further demonstrate the effectiveness of each component in our D-VQA on the IID dataset, we conduct the ablation studies on the VQA v2 [3] validation set, and present the experimental results in Table 1. From these results, all of the settings achieve a better performance than the original result of UpDn [2], which demonstrates the effectiveness of the components in our D-VQA.

Table 1: Effect of each component of our D-VQA on the model performance. We show the results on the VQA v2 dataset in terms of Accuracy (%). We use UpDn[†] [2] as the backbone bone.

| $(\tilde{v}_i, q_i)$ | $(v_i, \tilde{q}_i)$ | question-to-answer | vision-to-answer | VQA v2 validation (%) |
|---|---|---|---|---|
| | | | | 63.48 |
| √ | √ | | | 64.37 |
| | | √ | √ | **65.48** |
| √ | √ | √ | √ | 64.96 |

---

[*]Corresponding author

35th Conference on Neural Information Processing Systems (NeurIPS 2021).

**Algorithm 1** Training method of our D-VQA.

---

**Require:** Training data $\{(v_i, q_i, a_i)\}_{i=1}^{N}$, a base model $\mathcal{M}_b$, batch size $b$, threshold $\tau$.
1: Construct question-to-answer $\mathcal{M}_q$, vision-to-answer $\mathcal{M}_v$, and debiased branches $\mathcal{M}_d$.
2: Randomly initialise the parameters of $\mathcal{M}_{\{b,q,v,d\}}$.
3: **while** not converge **do**
4:     Randomly sample a mini-batch data $\{(v_i, q_i, a_i)\}_{i=1}^{b}$ from training data as positive samples.
5:     *// From the feature perspective*
6:     Forward $\mathcal{M}_q$, $\mathcal{M}_v$, and $\mathcal{M}_b$ with the positive samples as input, and then calculate $\mathcal{L}_{\{q,v,vqa\}}$.
7:     Obtain the bias detection weight $b_q$ and $b_v$ by Eq. (4).
8:     Obtain the negative biased features $\mathbf{f}_{bias}$ via Eq. (5).
9:     Forward the $\mathcal{M}_d$ with $\mathbf{f}_d = \mathbf{f} - \mathbf{f}_{bias}$ as input to calculate $\mathcal{L}_d$, and then obtain the debiased features $\mathbf{f}_c$ using Eq. (6).
10:     Take $(\mathbf{f}, \mathbf{f}_c, \mathbf{f}_{bias})$ as input to calculate the contrastive loss $\mathcal{L}_c$ by Eq. (7).
11:     *// From the sample perspective*
12:     **if** the training epoch higher than $\tau$ **then**
13:         Randomly sample images and questions from the mini-batch data $\{(v_i, q_i, a_i)\}_{i=1}^{b}$ to form two types of negative samples as $\{(\tilde{v}_i, q_i, a_i)\}_{i=1}^{b}$ and $\{(v_i, \tilde{q}_i, a_i)\}_{i=1}^{b}$
14:         Forward $\mathcal{M}_b$ with two types of negative samples as input, and obtain the sample perspective loss using Eq. (8).
15:     **end if**
16:     Update $\mathcal{M}_{\{b,q,v,d\}}$ by minimising the overall loss $\mathcal{L}$ (obtained via Eq. (9)).
17: **end while**

---

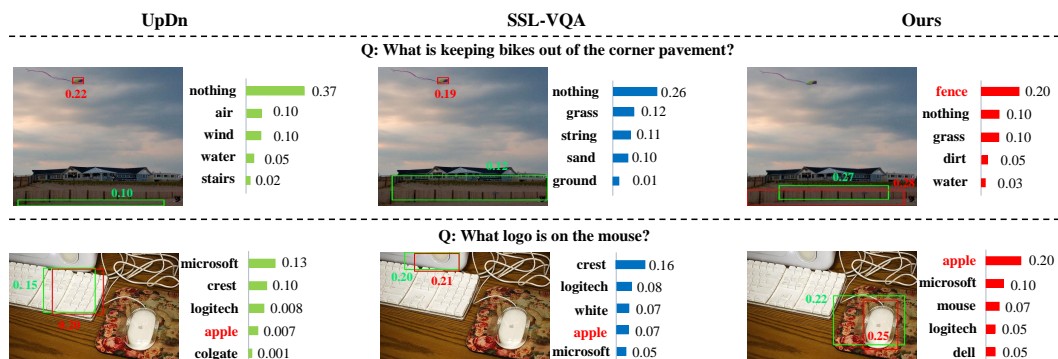

Figure 1: Qualitative comparison among UpDn [2], SSL-VQA [4], and our D-VQA on the VQA-CP v2 test set. For each example, we put the bounding box with top-2 attention weights in the image and show the answers with top-5 predictions. Red bold answer indicates the ground-truth answer.

Table 2: Language bias detection weights for the question types. Note that the higher weight it is, the stronger bias the question type has.

| Question Type | "How many" | "Is this" | "What color" | "what kind of" | "none of the above" |
|---|---|---|---|---|---|
| Language Bias Detection Weight | 0.40 | 0.73 | 0.33 | 0.22 | 0.51 |

## C   More visualisation results on VQA-CP v2

**Qualitative Comparison.** To further demonstrate the effectiveness of our D-VQA, we show more visualisation results in Figures 1 and 2. (1) In Figure 1, we display the predictions on the objects and answers of different methods (*i.e.,* UpDn [2], SSL-VQA [4], and our D-VQA). From these results, our D-VQA locates the target object accurately and outputs correct answers, while UpDn and SSL-VQA focus on other regions that are irrelevant to the questions and thus output wrong answers. (2) Besides, in Figure 2, we also visualise the answer distributions of different methods about different question types (*i.e.,* "How many . . . " and "Is this . . . "). From these results, the answer distributions between training and testing sets are very different. However, the backbone model UpDn captures the biases

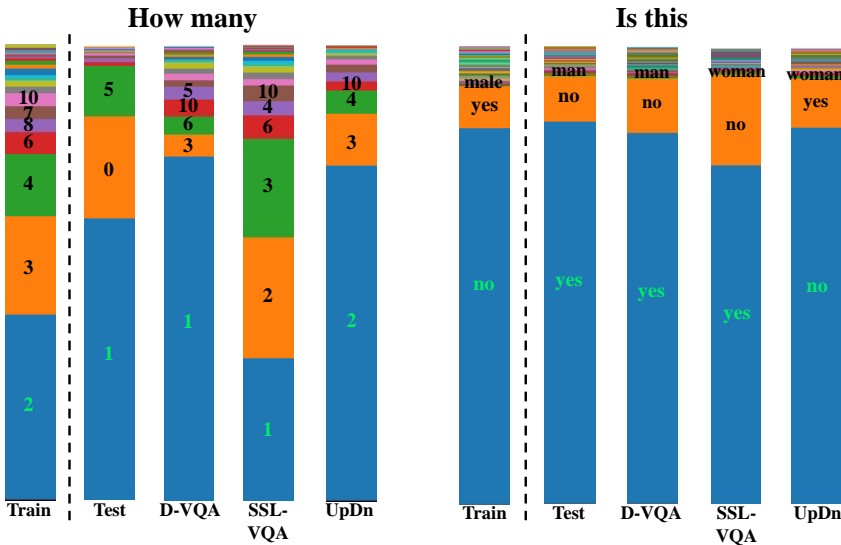

Figure 2: Qualitative comparison among UpDn [2], SSL-VQA [4] and our D-VQA on the VQA-CP v2 test set about the answer distributions.

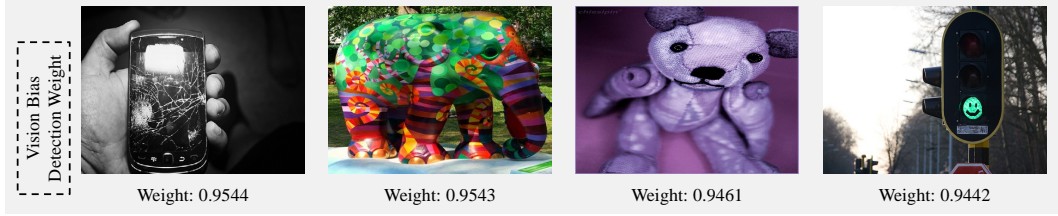

Figure 3: Examples of vision bias detection weights.

in the training set, and thus outputs a similar answer distribution with the training set when answering the questions in the test set, resulting in poor performance. Although SSL-VQA can alleviate the issue of biases, the effect is limited. Our D-VQA alleviates the negative effect of the language and vision biases from both feature and sample perspectives, and thus achieves similar answer distributions with the test set, embodying the better generalisation ability. These visualisation results demonstrate the effectiveness of our D-VQA.

**Visualisation of the bias detection modules.** To evaluate the language and vision bias detection modules qualitatively, we show the language and vision biases detection weights in Table 2 and Figure 3, respectively. Note that we obtain the average language bias detection weights by averaging all the bias detection weights $b_q$ of the questions with the same question type. Similarly, we gain the average vision bias detection weights via averaging all the bias detection weights $b_v$ of the samples with the same image. We have the following observations from these results: (1) As shown in Figure 2, the question type "How many . . ." has relative diverse answers but the answer "2" still dominant, and thus the average language bias detection weight is 0.4 (refer to Table 2). But in question type "Is this . . . ", the answer "no" occupies approximately 80% of all the answers, which has a strong bias, and thus the average language bias detection weight is 0.73 (refer to Table 2). (2) For the vision bias detection weights in Figure 3, each image contains one salient object that occupies approximately the whole image, and thus the vision bias detection weights have high values. These results demonstrate both the language and vision bias detection modules are able to recognise the corresponding true negative biases accurately, embodying the effectiveness of our bias detection modules.

**Analysis of the failure cases.** We demonstrate three typical failure cases of our D-VQA in Figure 4. From these results, we have the following observations. 1) As shown in (a) of the figure, our D-VQA pays attention to the appropriate location, and correctly predicts the colours of these dishes separately. However, since the ground-truth answer is not in the answer set that is constructed by us, our D-VQA

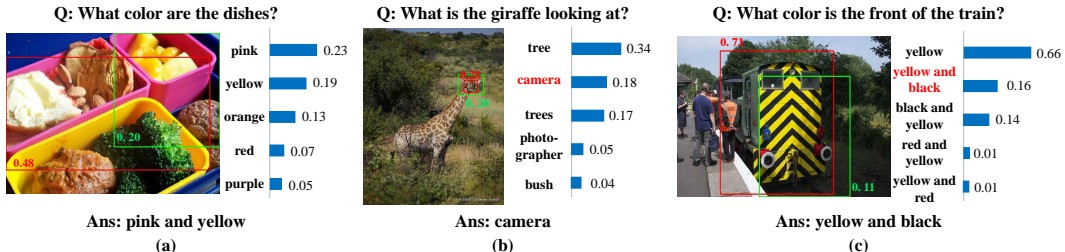

Figure 4: Typical failure cases of our D-VQA on the VQA-CP v2 test set.

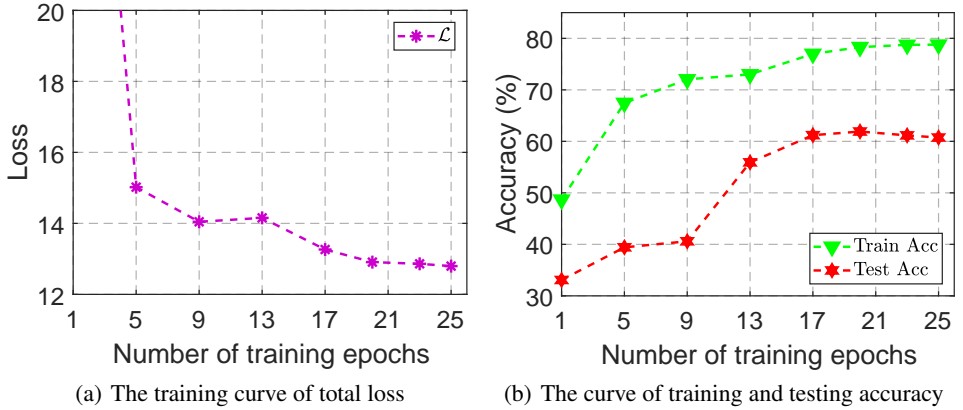

(a) The training curve of total loss    (b) The curve of training and testing accuracy

Figure 5: we show the training curve of the total loss and the curve of training and testing accuracy.

fails to answer this type of question. 2) In the example of (b), the giraffe in the image looks into the distance, it is hard to say what things are the giraffe looking on. However, the question is "What is the giraffe looking at?", the answer should be open-end. Nevertheless, our D-VQA also predicts the ground-truth answer "camera" at the second candidate. 3) Looking at the example of (c), our D-VQA predicts the answer excessively focus on the colour "yellow", and thus make true predictions at the second and third candidates.

**Visualisation of the training loss and the accuracy.** We show the training loss curve and the accuracy curve of our D-VQA in Figure 5. Note that we introduce the sample perspective loss at the 13-th epoch, and thus the training loss increases slightly. From these results, we have the following observations: (1) As the training epoch increases, the training loss converges gradually. (2) Meanwhile, the training and testing accuracy are also converged with the training process going on. These results demonstrate our D-VQA has a stable training procedure.