# OpenReview forum: "Debiased Visual Question Answering from Feature and Sample Perspectives"
_NeurIPS.cc/2021/Conference — NeurIPS 2021 Poster_

### Official Review · Reviewer_hQzS · 2021-07-09

**Rating:** 6
**Confidence:** 4

**Summary:**

This paper tackle explores how to partially annealed dataset biases when training VQA models.
 To do so, they applied to debiasing strategy:
  - they generate free negative samples to tentatively debased the answer distribution
 - they use a neural pipeline to tentatively debiased the multimodal features
Interestingly, both approaches are model-agnostic (up to $f = F(h,g)$), making the approach generic.

The paper has many merits and solid results. Multiple complementary experiments are insightful.
Yet, there are a few remaining ambiguities that prevent me from fully advocating for acceptance.
 I would be more than happy to increase my score if they are correctly addressed:

**Limitations And Societal Impact:**

LGTM

**Main Review:**



Key remarks:

A) The authors emphasize the debiasing aspect of the neural pipeline. It includes equations 3-7. My main concern is about equation 6 and the loss $L_d$.
First, can you elaborate on the $L_d$? What are the target labels of $L_d$ (or is it equivalent of $L_{vqa}$)? More generally, why this specific loss would only keep the “good” biases?
Second, as $f$ and $f_{bias}$ are trained independently (and thus, they do not live in the same space), why computing the difference would debias $f$?

Third, some design choices are not well-justified, e.g. why b_q.f_q instead of directly using $f_q$? Why using attention to combine $fˆ^ q_{bias}$ and $f^v_{bias}$? Even if I understand the intuition of those patterns, they remain arbitrary design choices that increase the complexity of the model. I would thus recommend the authors ablate them to justify them.

 B) In light of Table 2, the sampling procedure looks crucial. However, no ablation only contains feature-debiasing to determine its impact correctly. Therefore, it is hard to estimate whether feature debiasing has a marginal effect or not. Second, the negative samples are not correctly put in the perspective of the literature. It is mentioned that the process is inspired from 32 and 42 without more details. This is unfortunate as the current paper does show that sampling jointly questions and images is beneficial (which was not observed in 32 and 42). It took me quite some time to be sure that 3.3 was an actual (simple but valid) contribution to the paper.

 Potential improvements:

C) I understand the need to have $L_{sam}$ to remove the feature debasing pipelines. However, this aspect can be further study: 1) What happens if the loss is removed and the pipeline is removed at test time? Is there an implicit alignment of features or the debiasing signal is just lost? 2) What happens if the full pipeline is preserved at test time (more parameters), is the final higher as there is no proxy?

D) Positive biases are quite a controversial	topic; I would recommend detailing a bit more this section with the relevant literature. On the other side, there is no need to have such an extended related work section on self-supervised learning… as there is no self-supervised learning pipeline in the paper. Self-supervised learning is not CPC! Contrastive losses existed far before SimCLR!

Other remarks:
 - Overall, the paper is clearly written. I may recommend making figure 2, a bit bigger while showing the equation. Another idea would be to summarise all the equation 3-7 into a single one to help the global understanding of the model (in Appendix?).
Related work is correct. As mentioned, there is no need for detailed self-supervised learning literature.
L123: please add citations after “most previous methods.”
L218: are \hat{q} and \hat{v} from the same question-image pair?
Table1 is complete. Well appreciated
As mentioned in B, please complete the ablation in Table 2.
Tables 4 and 6 are persuasive.
Do you. Have some piece of intuition why D-VQA is so robust with few labels? Is it because of negative sampling (potential additional experiment)


Conclusion:
 This paper is consistent and has a pure-ML approach that I really appreciate. It also has solid results and convincing complementary experiments.
However, there are still some obscure points that I encourage the authors to clarify before I can recommend acceptance.
Besides, if some of the experiments I recommended were performed (e.g, Complete Tab2, ablate $L_sim$, ablate neural choice), I am eager to increase my score even more.


**Time Spent Reviewing:**

4h

---

> ### Author Response · Authors · 2021-08-10
> **Authors' response**
>
> **Q1.** My main concern is about equation 6 and the loss $L_d$. First, can you elaborate on the $L_d$? What are the target labels of $L_d$ (or is it equivalent of $L_{vqa}$)? More generally, why this specific loss would only keep the “good” biases?
>
> **A1.** $L_d$ is the binary cross-entropy loss of the debiasing branch, whose target label is the ground truth label in the dataset (the target label is the same as $L_{vqa}$).
>
> The debiasing branch seeks to obtain the debiased features $f_c$ to perform visual question answering. In this sense, $L_d$ is able to guide the bias detection module to recognize and seek to alleviate the negative biases as well as possible.
>
> We do not claim that the $L_d$ would only keep the ''good'' biases. Instead, we apply $L_d$ to guide the debiasing branch to obtain the features that reduce the negative biases as well as possible.
>
> ---
>
> **Q2.** As $f$ an $f_{bias}$ are trained independently (and thus, they do not live in the same space), why computing the difference would debias $f$?
>
> **A2.** In the debiasing branch, we seek to perform the VQA task with the debiased features. Specifically, the debiased features are obtained by removing negative biased features $f_{bias}$ from the multimodal feature $f$, namely, $f_d = f - f_{bias}$.
>
> As shown in Line 4-6 in Table 2 of the paper, when adding question-to-answer and vision-to-answer branches, the performance further improves. The results demonstrate these two branches are able to capture the negative biases to be removed.
>
> ---
>
> **Q3.** Some design choices are not well-justified, e.g. why $b_q \cdot f_q$ instead of directly using $f_q$? (why $b_q \cdot f_q$ instead of directly using $f_q$?)
>
> **A3.** The feature $f_q$ is the output feature from the question-to-answer branch, where $f_q$ contains the language biases. In this paper,  we argue that the biases can be positive or negative. For example, for the positive bias, the question type ("What colour ...") will narrow the range of the answers; For the negative bias, the models tend to capture superficial correlations between one modal information and the answers. In this sense, we explicitly devise the bias detection modules (with output as $b_q$) to detect and seek to alleviate the negative biases. In this way, we adopt $b_q \cdot f_q$ to denote the negative biased features that should be reduced.
>
> In Table 3 of the paper, we show the ablation studies over the bias detection modules. From the results, both the language and vision bias detection modules promote the model performance, which demonstrates the effectiveness of the bias detection modules. Moreover, combining two bias detection modules together would further promote the model performance, which further demonstrates the importance of debiasing in both language and vision modalities.
>
> ---
>
> **Q4.** Why using attention to combine $f_{bias}^q$ and $f_{bias}^v$?
>
> **A4.** For one question-image pair, it is non-trivial on judging which types of biases are dominant. To achieve this, we adopt an attention mechanism to obtain the weights for different biases to fuse these two negative biased features
>
> We also conduct an experiment on VQA-CP v2, in which we replace the attention mechanism by directly adding $f_{bias}^q$ and $f_{bias}^v$. We show the results below:
>
> |Model| VQA-CP v2 test (%)|
> |:---:|:-----:|
> |D-VQA w/o attn | 60.23 |
> |D-VQA| 61.91 |
>
> Our D-VQA performs better than itself without the attention mechanism, which demonstrates the effectiveness and necessity of the attention mechanism on fusing these two negative biased features.
>
> ---
>
> **Q5.** However, no ablation only contains feature-debiasing to determine its impact correctly. Therefore, it is hard to estimate whether feature debiasing has a marginal effect or not.
>
> **A5.** In this paper, we focus on debiasing issues from both the sample perspective and feature perspective. Indeed, when using feature debiasing only, the improvement is marginal, but the debiasing on feature perspective is still very important. In our primary experiments, we find that if the bias issue on the sample perspective is not well addressed, the debiasing on feature perspective alone may not work very well. Combining these two debiasing perspectives would further improve the performance (Refer to the ablation studies in Table 2 of the paper).
>
> ---
>
> **Q6.** Second, the negative samples are not correctly put in the perspective of the literature.
>
> **A6.** Thanks for your valuable suggestions. We will carefully check up the literature about the negative samples in the paper.
>
> ---
>
> **Q7.** I understand the need to have $L_{sam}$ to remove the feature debasing pipelines. However, this aspect can be further study: 1) What happens if the loss is removed and the pipeline is removed at test time? Is there an implicit alignment of features or the debiasing signal is just lost?
>
> **A7.** Removing the $L_{sam}$ means removing the sample perspective debiasing pipeline, and remain the feature perspective pipeline only. At test time, only using the feature perspective pipeline performs worse than considering two perspectives. For more details please refer to the answer to **Q5**.
>
> Besides, in the feature perspective, we adopt contrastive loss to impel the multimodal feature to close to the debiased features as well as possible and meanwhile to keep away from the negative biased features, to achieve the target of debiasing.
>
> ---
>
> **Q8.** What happens if the full pipeline is preserved at test time (more parameters), is the final higher as there is no proxy?
>
> **A8.** When preserving the full pipeline at test time, our method achieves better performance (66.59% *vs.* 61.91%).
>
> ---
>
> **Q9.** Positive biases are quite a controversial topic; I would recommend detailing a bit more this section with the relevant literature.
>
> **A9.** The positive bias has been proposed by [a,b] in Visual Question Answering. Here, the positive biases in the datasets represent natural properties of the real world, which is usually helpful for the generalisation performance. For example, the question type ("What colour ...") will narrow the range of the answers. We will carefully revise it in the paper for clearer.
>
> ---
>
> **Q10.** there is no need to have such an extended related work section on self-supervised learning… as there is no self-supervised learning pipeline in the paper. Self-supervised learning is not CPC! Contrastive losses existed far before SimCLR!
>
> **A10.** Thanks for your valuable suggestions. We will carefully revise the related work in this paper.
>
> ---
>
> **Q11.** I may recommend making figure 2, a bit bigger while showing the equation. Another idea would be to summarise all the equations 3-7 into a single one to help the global understanding of the model (in Appendix?). L123: please add citations after "most previous methods"
>
> **A11.** Thanks for your valuable suggestions. We will carefully revise Figure 2 and the summarization of the equation 3-7 in this paper. We will add citations after "most previous methods" in L123, such as [c,d,e].
>
> ---
>
> **Q12.** L218: are $\hat{q}$ and $\hat{v}$ from the same question-image pair?
>
> **A12.** $\hat{q}$ and $\hat{v}$ have a probability that coming from the same question-image pair. Specifically, in a mini-batch data $(v\_i, q\_i, a\_i)\_{i=1}^n$, we construct negative samples for the instance $(v\_j, q\_j, a\_j)$ by randomly sampling one question $\hat{q}\_j$ in the mini-batch data $(q\_i)\_{i=1}^n$ to form the negative samples $(v\_j, \hat{q}\_j, a\_j)$. Similary, another negative sample $(\hat{v}\_j, q\_j, a\_j)$ is formed in the same way.
>
> ---
>
> **Q13.** Do you have some piece of intuition why D-VQA is so robust with few labels? Is it because of negative sampling?
>
> **A13.** The negative sampling is helpful for the robustness of D-VQA, since our sample perspective strategy makes full use of the samples available. Meanwhile, the contributions of the feature perspective strategy cannot be ignored.
>
> To evaluate the robustness of the D-VQA on little labelled data, we conduct ablation studies of the sample perspective on the different proportions of the training data, and the experimental results are in the table below:
>
> |Model| 20%|40%|60%|80%|100%|
> |:------:|:------:|:------:|:------:|:------:|:-----:|
> |SSL-VQA|52.71|54.42|56.83|57.31|57.59|
> |D-VQA with sample perspective only| 52.83| 55.64| 57.61| 58.32| 60.24|
> |D-VQA| 52.94|56.74|58.31|59.05|61.91|
>
> From the results, D-VQA with sample perspective only achieves better performance, while combining these two perspectives would further improve the performance.
>
> ---
>
> **Reference:**
>
> [a]   Counterfactual VQA: A cause-effect look at language bias. In CVPR.
>
> [b]   MUTANT: A training paradigm for out-of-distribution generalization in visual question answering. In EMNLP.
>
> [c]   Rubi: Reducing unimodal biases for visual question answering. In NeurIPS.
>
> [d]   Overcoming language priors in visual question answering with adversarial regularization. In NeurIPS.
>
> [e]   Overcoming language priors with self-supervised learning for visual question answering. In IJCAI.

---

### Official Review · Reviewer_jcD5 · 2021-07-12

**Rating:** 7
**Confidence:** 4

**Summary:**

The paper introduces “D-VQA”, a training approach dedicated to remove harmful biases (both language and vision ones) in VQA models. D-VQA performs in two ways, namely: feature-wise and sampling-wise.

On the features side, during its training, D-VQA attempts to de-bias the features learned by a VQA model by subtracting the (deliberately) biased features learned by unimodal language-only and vision-only models. Moreover, in order to get rid of the unimodal branches during the inference, D-VQA training includes an additional contrastive loss which aims to bring the original VQA features closer to the debiased ones (i.e. the ones obtained after feature subtraction with the unimodal branches).

On the sampling side, D-VQA employs an existing approach proposed in [42] (which is properly mentioned and recognized by the authors) introducing negative sampling triplets for each positive triplet (by randomly changing either an image or a question in a positive triplet).

Combining the 2 aforementioned techniques allows D-VQA to obtain the state-of-the-art results on both VQA-CP and VQAv2 comparing to the existing bias-reduction methods (while using the UpDn model as a backbone). In addition to that, the authors demonstrate that their approach is model-agnostic (for example, it is compatible with a more recent and powerful LXMERT backbone).

**Limitations And Societal Impact:**

The "broader impact" statement has been included in the article and seems reasonable.

**Main Review:**

*Originality*

The paper rather exploits and successfully combines already-existing ideas for bias-reduction in VQA (by clearly citing the original works) than proposes radically new ones.

Thus, the sampling of negative triplets was proposed in [42]. In this work, the authors straightforwardly extend this idea by randomly switching both images and questions.

Similarly, de-biasing the VQA features via learning deliberately biased unimodal ones was also proposed in many recent papers (for example, in [9]).

In my opinion, the main originality of the present work is the idea of the contrastive learning (the loss L_c) which makes the VQA model independent of its unimodal branches at the inference stage (and, therefore, accelerates and simplifies the final model).

*Quality*

To the best of my knowledge, the significant related work has been properly cited and compared with the proposed approach. The claimed contributions are supported with conclusive experimental results and ablation studies on the VQA-CP and VQAv2 datasets.

*Clarity*

The paper is clearly written and easy to follow. Some important implementation details can be found in Supplementary Materials.

*Significance*

All in all, D-VQA can be considered as a significant contribution as it demonstrates the complementarity of two (almost orthogonal) branches of the research on bias reduction in VQA, namely: feature de-biasing and training balancing.

The results could have been even more convincing had the authors tested their approach on other unbiased datasets than VQA-CP (for example, GQA-OOD: https://github.com/gqa-ood/GQA-OOD - the dataset has been published after the NeurIPS deadline so the lack of evaluation on it cannot be considered as a problem).

*Minor remarks / questions*

1.	In the ablation study reported in Table 3, you demonstrate that amplifying the unimodal biased feature vectors f_q and f_v with the estimated real-values b_q and b_v (obtained via “bias detection modules”) improves the performances. Would not it work even better if b_q and b_v were vectors (as in [9], for example) rather than simple real values (in this case, f^{bias}_q could be a pointwise product between f_q and b_q)?

2.	I am not sure to understand the Equation 5. Does “a” denote a vector or a single real value? Indeed, on the one hand, you state that “m_a” outputs a single real value. But on the other hand, “a” is an output of the softmax operation and is also transposed in Eq. 5 indicating that it is rather a vector.

*Typos*

1.	Line 103 (and also in other parts of the article): you employ the word “auxiliary” as a verb. I am not a native speaker, but I do not understand what that means, and I am not sure that it is grammatically correct.

2.	Line 195: “contains” -> “containing”.



**Time Spent Reviewing:**

7

---

> ### Author Response · Authors · 2021-08-10
> **Authors' response**
>
> **Q1.** The sampling of negative triplets was proposed in [42]. In this work, the authors straightforwardly extend this idea by randomly switching both images and questions.
>
> **A1.** In this paper, we consider the biases in both language and vision modalities, and hope to make full use of the data available to improve the sensitiveness of the model in both the image and questions. Based on this intuition, we construct two types of negative samples (negative question or negative image) to assist the training of the VQA model.
>
> In [42] (SSL-VQA[a]), they only consider the language bias, and seek to construct the negative sample to alleviate the language bias.
>
> ---
>
> **Q2.** Similarly, de-biasing the VQA features via learning deliberately biased unimodal ones was also proposed in many recent papers (for example, in [9]).
>
> **A2.** In our D-VQA, we consider the biases in both language and vision modalities and build question-to-answer and vision-to-answer branches to capture the language and vision biases, respectively. we argue that the biases can be positive or negative. For example, for the positive bias, the question type ("What colour ...") will narrow the range of the answers; For the negative bias, the models tend to capture superficial correlations between one modal information and the answers. Thus, we explicitly construct the bias detection modules to recognise and seek to alleviate the negative biases.
>
> In [9] (RUBI[b]), they consider language bias only and construct a question-only branch to assist in weakening the language bias.
>
> Although both RUBI[b] and our D-VQA introduce the additional branch, the usages are different. RUBI[b] seeks to adjust the weight of the samples based on the additional branch, while our D-VQA seeks to obtain debiased features to achieve better performance.
>
> ---
> **Q3.**  In my opinion, the main originality of the present work is the idea of the contrastive learning (the loss $L_c$) which makes the VQA model independent of its unimodal branches at the inference stage (and, therefore, accelerates and simplifies the final model)
>
> **A3.** The idea of contrastive learning is one of our novelty, more novelty can be summarised as follows:
>
> We consider the biases in both language and vision modalities, while most previous methods consider the language bias only.
>
> We realize that not all biases are harmful to the models, and explicitly construct two bias detection modules to recognise and seek to alleviate the negative biases.
>
> We construct two types of negative samples (negative question or image) to increase the sensitiveness of the model on both language and vision modalities.
>
> ---
>
> **Q4.** The results could have been even more convincing had the authors tested their approach on other unbiased datasets than VQA-CP (for example, GQA-OOD: https://github.com/gqa-ood/GQA-OOD (https://github.com/gqaood/GQA-OOD) - the dataset has been published after the NeurIPS deadline so the lack of evaluation on it cannot be considered as a problem).
>
> **A4.** We appreciate the reviewer for providing the new benchmark dataset GQA-OOD[c] for us. Since GQA-OOD has been published after the NeurIPS deadline, we cannot evaluate our D-VQA on it.
>
> In the rebuttal period, we evaluate our D-VQA on GQA-OOD, and the results are shown in the table below:
>
> |Method| acc-all | acc-tail| acc-head|
> |:------|:------:|:------:|:------:|
> |BUTD|46.4|42.1|49.1|
> |+BP[d]|33.1|30.8|34.5|
> |+LM[d]|34.5|32.2|35.9|
> |+RUBI[b]|38.8|35.7|40.8|
> |+RUBI+QB[b]|46.7|42.1|49.4|
> |+D-VQA(Ours)|49.3|44.4|52.7|
>
> From these results, our D-VQA achieves the best performance, which demonstrates the superiority of our D-VQA.
>
> ---
>
> **Q5.** In the ablation study reported in Table 3, you demonstrate that amplifying the unimodal biased feature vectors $f_q$ and $f_v$ with the estimated real-values $b_q$ and $b_v$ (obtained via “bias detection modules”) improves the performances. Would not it work even better if $b_q$ and $b_v$ were vectors (as in [9], for example) rather than simple real values (in this case, $f^{bias}_q$ could be a pointwise product between $f_q$ and $b_q$).
>
> **A5.** Thank you very much for your valuable comments. In fact, we have tried to set $b_q$ and $b_v$ as vectors during the experiments. However, we found that it performs slightly worse than setting  $b_q$ and $b_v$ as values (60.55% *vs.* 61.91%).
>
>  In our setting, we argue that setting $b_q$ and $b_v$ to be values may be more suitable. Specifically, as we need to construct the bias detection modules to recognize the negative biases, the value representation can intuitively show us the extent of negative bias in the whole question or image.
>
> Furthermore, in Table 1 and Figure 2 in the supplementary material, we qualitatively demonstrate that our bias detection modules are able to recognize the negative biases accurately. In this sense, setting $b_q$ and $b_v$ to be values would be more interpretation-friendly.
>
> ---
> **Q6.**  I am not sure to understand the Equation 5. Does “$a$” denote a vector or a single real value? Indeed, on the one hand, you state that “$m_a$” outputs a single real value. But on the other hand, “$a$” is an output of the softmax operation and is also transposed in Eq. 5 indicating that it is rather a vector.
>
> **A6.** Equation 5 is to fuse these two negative biased features using attention mechanism. Specifically, in equation 5, $a \in R^{2}$ denote a vector. $f $ = [$f_1$; $f_2$] denotes stacking these two features, where $f \in R^{2 \times d}$. Moreover, "$m_a$" denotes an MLP, which transforms the stacked features at the last dimension ($R^{2 \times d} \rightarrow R^{2}$ ). we will carefully revise the equation to make it clearer.
>
> ---
>
> **Q7.** Typos
>
> **A7.**  Thanks for your valuable suggestions, we will carefully revise the grammar in this paper.
>
> ---
>
> **Reference:**
>
> [a]   Overcoming language priors with self-supervised learning for visual question answering. In IJCAI.
>
> [b]   Rubi: Reducing unimodal biases for visual question answering. In NeurIPS.
>
> [c]   Roses Are Red, Violets Are Blue... but Should Vqa Expect Them To? In CVPR.
>
> [d]   Don’t take the easy way out: Ensemble based methods for avoiding known dataset biases. In EMNLP.

---

### Official Review · Reviewer_e8gV · 2021-07-13

**Rating:** 7
**Confidence:** 4

**Summary:**

This work addresses a known bias in visual question answering tasks, in which networks learn to pick an answer based on a single modality without perceiving the entire input.  The work novelty comes from proposing a set of regularization strategies s.t: (1) reducing "bad" bias that is not a useful commonsense knowledge;  (2) reducing computational inferences through contrastive loss; (3) considering an image-only bias; (4) employ negative sampling to increase attention sensitiveness on both modalities. In both VQA and VQA-CP, the approach shows state-of-the-art performance.

**Limitations And Societal Impact:**

The authors have adequately addressed the limitations and potential negative social impact of their work

**Main Review:**

Originality: A known bias with VQA models is that they do not perceive all inputs and take shortcuts. There have been various attempts to fix this issue through training an additional unimodal classifier and removing those biases (e.g., RUBi, LMH). However, this work offers some novel ideas, like negative sampling and contrastive loss. Further, the proposed approach allows the choice of unimodal features that are not contributing to the joint task, while previous works argue that unimodal features should be completely removed.

Quality: Most of the techniques used are conventional (e.g., negative sampling, contrastive loss). Therefore, I consider the approach to be technically sound.  I recommend providing more qualitative examples, particularly negative ones. Also, in my view, since the bias-detection module is a novelty, every supporting evidence for bias-detection in the supplementary material (e.g., Tab 1, Fig 2) is appropriate for the main paper.

Clarity: The submission is clear, and the ideas are intuitive. The appendix provides implementation details but no code. A point I find quite confusing is the bias detection model's ability to distinguish between a 'good' and a 'bad' bias. A 'good' bias, according to the authors, helps the joint task. However, the joint task does pick a 'bad' bias as well. I would appreciate a clarification on this.

Significance: Visual question answering is an important and popular task. The bias discussed in this paper is a major concern in any VQA model, which is why the popular VQA-CP dataset was developed.  The proposed techniques should aid in advancing the goals of answering questions using reasoning instead of unaware biased correlations.  Importantly,  the experiments show a significant improvement in the performance of the VQA-CP task (61.91 vs. 57.59)

To summarize, here's what concerns me:
[W1] Are there any negative effects for the proposed methods?
[W2] Can the authors comment on the impact of different f_q/f_v? For instance, f_g can be an LSTM encoder for textual input.
[W3] Why the b_q layer picks only 'bad' bias?



**Time Spent Reviewing:**

6

---

> ### Author Response · Authors · 2021-08-10
> **Authors' response**
>
> **Q1.** The work novelty comes from proposing a set of regularization strategies s.t: (1) reducing "bad" bias that is not a useful commonsense knowledge; (2) reducing computational inferences through contrastive loss; (3) considering an image-only bias; (4) employ negative sampling to increase attention sensitiveness on both modalities. In both VQA and VQA-CP, the approach shows state-of-the-art performance.
>
> **A1.**  Thank you very much for your recognition and summary of our novelty and contributions.
>
> ---
>
> **Q2.**  I recommend providing more qualitative examples, particularly negative ones.
>
> **A2.** Thank you for your valuable suggestions. We will provide more qualitative examples (especially for the failure cases of our methods) in the main paper and supplementary material.
>
> ---
>
> **Q3.** The submission is clear, and the ideas are intuitive. The appendix provides implementation details but no code.
>
> **A3.** Thank you very much for your suggestion. We will release our code upon acceptance.
>
> ---
> **Q4.** A point I find quite confusing is the bias detection model's ability to distinguish between a 'good' and a 'bad' bias. A 'good' bias, according to the authors, helps the joint task. However, the joint task does pick a 'bad' bias as well. I would appreciate a clarification on this.
>
> **A4.** The joint task indeed contains “bad” bias. To alleviate this issue, we explicitly devise the bias detection modules to filter and then seek to alleviate the negative biases. To this end, we devise the loss $L_d$ of the debiasing branch to guide the bias detection module. Please refer to the answer to **Q7** for more details.
>
> ---
> **Q5.** Are there any negative effects for the proposed methods?
>
> **A5.** Thank you very much for your question. Our method may have the following limitations.
>
> **First**, we currently focus on the visual question answering task only. In the future, we will extend our debiasing technique to other vision-and-language tasks, such as Referring Expression Comprehension.
>
> **Second**, our D-VQA achieves state-of-the-art performance on most types of questions (See Table 1 of the paper), but there is a larger room to improve on questions like 'Num' and 'Other'.
>
> ---
>
> **Q6.** Can the authors comment on the impact of different $f_q$/$f_v$? For instance, $f_g$ can be an LSTM encoder for textual input.
>
> **A6.** For a fair comparison, we use the same object feature extractors (*i.e.* Faster-RCNN) with classical works[a,b,c] to extract the object features. But for the question feature extractor, when taking the SAN, UpDn, and LXMERT as the backbone, they adopt LSTM, GRU, and BERT to capture the question features, respectively.
>
> From the results in Table 4 of the paper, all the backbones + D-VQA achieve better performance, demonstrating that not only the basic model (LSTM/GRU) but also the bigger one (BERT) can capture the bias to tackle the tasks.
>
> ---
>
> **Q7.** Why the $b_q$ layer picks only 'bad' bias?
>
> **A7.** We explicitly devise the bias detection modules ($b_q$) to filter and then seek to alleviate the negative biases. In the debiasing branch, based on the output of $b_q$, we obtain the debiased features to perform the VQA task. Moreover, the debiasing branch (including bias detection modules) is trained by binary cross-entropy loss $L_d$. Here, the target label is the same as the $L_{vqa}$. With the guidance of the $L_d$, the bias detection modules are expected to recognise the true negative biases.
>
> Quantitative (Table 3 in the paper) and qualitative results (in the supplementary material of the paper) demonstrate our bias detection modules are able to capture the negative biases accurately, embodying the effectiveness of our bias detection modules.
>
> ---
>
> **Reference:**
>
> [a] Rubi: Reducing unimodal biases for visual question answering. In NeurIPS.
>
> [b] Overcoming language priors with self-supervised learning for visual question answering. In IJCAI.
>
> [c] Counterfactual VQR:A cause-effect look at language bias. In CVPR.

---

### Official Review · Reviewer_pgR5 · 2021-07-20

**Rating:** 6
**Confidence:** 4

**Summary:**

The paper targets the bias problem in visual question answering task where the models may only capture the biases instead of showing real reasoning abilities. The paper introduces a method called D-VQA which tries to alleviate this problem from feature and sample perspective. With two unimodal bias detection modules, D-VQA recognizes and removes the negative biases while authors also construct two types of negative samples to increase sensitiveness of VQA models to both modalities by minimizing the possibility of predicting correct ground-truth answer of the positive sample. The results on VQA-CP and VQA v2 dataset show the effectiveness of the approach.

**Limitations And Societal Impact:**

The broader impact statement has been included and is reasonable.

**Main Review:**

The paper introduces an approach called D-VQA which targets the bias problem in VQA. The paper is well written and is easy to follow and targets an important task which can help alleviate problems in the data collection methods such as bias that can occur naturally. My comments follow below:

1) For feature based debiasing, the model has two unimodal bias detection modules which pass image alone and text alone through the model to obtain the bias feature. This bias feature is removed from the multimodal feature and optimized for binary cross entropy loss for answers. The original multimodal feature is brought close to bias free multimodal feature through contrastive loss making sure that no extra parameters are used during inference.
2) For sample based debiasing, the authors sample negative sample for positive pairs by replacing either the text or the image. The binary cross entropy loss for the negative pairs is maximized so that the model is sensitive to the multimodal input and not focus on bias in the unimodal input.
3) For the ablations, they should have also been conducted on VQA v2 so that reader can understand the impact and differences on the well-calibrated benchmark. At the moment, it is hard to say how exactly the ablations work in a setting with actual bias.
4) For the LXMERT+D-VQA option, it would have been useful to have numbers on VQA v2 to understand if D-VQA introduces a drop or gain in performance for the models which are already performing very high on VQA v2 task.
5) The crucial details for the training have been added into the supplementary but instead should be including in the main paper as the paper feels incomplete without them. Also, is the LXMERT backbone pretrained or not? If yes, how does D-VQA affect pretraining?
6) From the ablations, it looks like that the sample based debiasing on vision side results in the most gain on the final results. There is no independent evaluation of just using feature based "question-to-answer" and "vision-to-answer" debiasing and understand how impactful those are. This is in contrast with how paper is structured mostly around feature-based debiasing even though sample-based debiasing which provides the most gain is only discussed for one paragraph. This questions the novelty of the paper as the sample based debiasing on vision side is obvious given numerous past study on the question side bias and is simple to add.
7) Table 2, Row 1 is technically simple UpDn. Why is that number different compared to Table 1, Row 3?
8) Authors stress on the point that they don't use any extra data for their augmentations. But contrary, why not improve results possibly by using techniques from MUTANT etc. It would have been good to see if extra data augmentations techniques can work well with DVQA.
9) Figure 1 necessarily doesn't show the problem of bias in VQA clearly and probably should something more concrete along with questions.
10) The name D-VQA is too general and I would suggest authors to rename title of their paper to be something more specific as it doesn't completely solved the problem of bias in VQA as it currently seems like reading the paper title.
11) The paper only tests the debiasing on images from COCO domain specifically. It would be good to see generalization to other domains such as TextVQA as well.


Post Rebuttal:
Please check my response to the authors.


**Time Spent Reviewing:**

6

---

> ### Author Response · Authors · 2021-08-10
> **Authors' response**
>
> **Q1.** For the ablations, they should have also been conducted on VQA v2 so that reader can understand the impact and differences on the well-calibrated benchmark. At the moment, it is hard to say how exactly the ablations work in a setting with actual bias.
>
> **A1.**  Thank you very much for your valuable suggestions.
>
> **First**, in this paper, following existing works [a, b, c], we evaluated our methods on both VQA v2 and VQA-CP v2 (See results in Table 1 of the paper). Moreover, considering that existing debiased methods [a, b, c] often perform ablation studies on VQA-CP v2, we also evaluate the reasoning ability of our method on VQA-CP v2.
>
> **Second**, to respond the Reviewer's concerns, we also perform ablation studies on VQA v2, and report the results below:
>
> | $(\hat{v}_i, q_i)$ | $(v_i, \hat{q}_i)$ | question-to-answer | vision-to-answer | VQA v2 val (%)|
> |:-----:|:-----:|:-----:|:-----:|:-----:|
> |$\surd$|$\surd$| - | - | 64.37 |
> | -| - | $\surd$ |$\surd$| 65.48 |
> |$\surd$|$\surd$|$\surd$|$\surd$| 64.96 |
>
> From the ablation results above, our method performs better than the baseline UpDn (See Table 1 of the paper). This further demonstrates the effectiveness of the proposed modules. It is worth noticing that similar trend of ablation results can be also observed on VQA-CP v2 in the original submission (See Table 2 in the paper).
>
> ---
>
> **Q2.** For the LXMERT+D-VQA option, it would have been useful to have numbers on VQA v2 to understand if D-VQA introduces a drop or gain in performance for the models which are already performing very high on VQA v2 task.
>
> **A2.**  To demonstrate the effectiveness of our method, we conduct experiments on VQA v2 validation split based on the backbone LXMERT, and show the results in the table below.
>
> |Model| All | Yes/No | Num | Other|
> |:---:|:----:|:----:|:----:|:----:|
> |LXMERT| 74.16 | 89.31 | 56.85 | 65.14 |
> |LXMERT + D-VQA(Ours)| 78.84 | 94.04 | 66.38 | 70.56 |
>
> From the results, LXMERT + D-VQA achieves better performance, which demonstrates the effectiveness of our method.
>
> ---
>
> **Q3.** The crucial details for the training have been added into the supplementary but instead should be including in the main paper as the paper feels incomplete without them. Also, is the LXMERT backbone pretrained or not? If yes, how does D-VQA affect pretraining?
>
> **A3.**  Thanks for your valuable suggestions, and we will carefully revise the main paper to make the training details clearer. Actually, we do not perform pre-training on LXMERT with our D-VQA. Instead, we load the pre-trained LXMERT model from the official GitHub repository, and then perform downstream VQA-CP task with our D-VQA method. Moreover, we will release our code upon acceptance.
>
> ---
>
> **Q4.** From the ablations, it looks like that the sample based debiasing on vision side results in the most gain on the final results. There is no independent evaluation of just using feature based "question-to-answer" and "vision-to-answer" debiasing and understand how impactful those are. This is in contrast with how paper is structured mostly around feature-based debiasing even though sample-based debiasing which provides the most gain is only discussed for one paragraph. This questions the novelty of the paper as the sample based debiasing on vision side is obvious given numerous past study on the question side bias and is simple to add.
>
> **A4.** In this paper, we focus on debiasing issues from both the sample perspective and feature perspective. We agree that the debiasing on sample perspective provides more performance gain from the experiments, which is consistent with past studies.
>
> However, in our paper, we argue that the biases can be positive or negative. For example, for the positive bias, the question type ("What colour ...") will narrow the range of the answers; For the negative bias, the models tend to capture superficial correlations between one modal information and the answers. In this paper, we explicitly devise the bias detection modules to detect and seek to alleviate the negative biases.
>
> Here, we argue that if the bias issue on the sample perspective is not well addressed, the debiasing on feature perspective alone may not work very well. Nevertheless, the debiasing on feature perspective is also important. From the ablation studies in Table 2 of the paper, considering two perspectives of debiasing can achieve better performance.
>
> ---
>
> **Q5.** Table 2, Row 1 is technically simple UpDn. Why is that number different compared to Table 1, Row 3?
>
> **A5.** The UpDn results in Table 1 and Table 2 indeed are slightly different due to the following reasons. In Table 1, for simplicity, we directly use the results from the paper [d] for UpDn. In Table 2, following SSL-VQA [b], we re-implement UpDn as our backbone, which performs better than the original version (in Table 1). In the revised paper, we will carefully explain the differences.
>
> ---
>
> **Q6.** Authors stress on the point that they don't use any extra data for their augmentations. But contrary, why not improve results possibly by using techniques from MUTANT etc. It would have been good to see if extra data augmentations techniques can work well with D-VQA.
>
> **A6.** In the current paper, we do not focus on improving the performance by simply using extra data. In contrast, we observe the bias issue in VQA and seek to alleviate the bias issue by making full use of the data available.
>
> Extra data often benefit the learning performance. However, data augmentation techniques (such as MUTANT[e]) often require additional annotations, which could be very expensive. In this sense, it may be non-trivial to apply such techniques in more general cases.
>
> In the future, we will consider augmenting the data to further improve learning performance.
>
> ---
>
> **Q7.** Figure 1 necessarily doesn't show the problem of bias in VQA clearly and probably should something more concrete along with questions.
>
> **A7.** In Figure 1, we seek to demonstrate the vision bias only, where the answers usually corresponding to the most salient objects/attributes in the images. To respond to the Reviewer's concerns, we will carefully revise Figure 1 to better demonstrate the dataset bias.
>
> ---
>
> **Q8.**  The name D-VQA is too general and I would suggest authors to rename title of their paper to be something more specific as it doesn't completely solved the problem of bias in VQA as it currently seems like reading the paper title.
>
> **A8.** Thanks for your valuable suggestions, and we will carefully revise the title of the paper to make the title more concretely.
>
> ---
>
> **Q9.**  The paper only tests the debiasing on images from COCO domain specifically. It would be good to see generalization to other domains such as TextVQA as well.
>
> **A9.**   Thanks for your valuable suggestion. To further study the generalization ability of our method, we will extend our method in TextVQA in future work. In this paper, we follow classical works[a,b,c] of reducing bias, and conduct experiments on VQA-CP v2 and achieve state-of-the-art performance.
>
> In the rebuttal period, due to the time limit, we hard to conduct complete experiments on the TextVQA that requires OCR module to recognise tokens in images. We agree that it is essential to study the generalisation problem. In this sense, we conduct experiments on a new benchmark dataset GQA-OOD[f]. We report the results below:
>
> |Method| acc-all | acc-tail| acc-head|
> |:------|:------:|:------:|:------:|
> |BUTD|46.4|42.1|49.1|
> |+BP[d]|33.1|30.8|34.5|
> |+LM[d]|34.5|32.2|35.9|
> |+RUBI[b]|38.8|35.7|40.8|
> |+RUBI+QB[b]|46.7|42.1|49.4|
> |+D-VQA(Ours)|49.3|44.4|52.7|
>
> From these results, our method achieves the best performance, which demonstrates the better generalization ability of our method.
>
> ---
>
> **References:**
>
> [a]    Rubi: Reducing unimodal biases for visual question answering. In NeurIPS.
>
> [b]    Overcoming language priors with self-supervised learning for visual question answering. In IJCAI.
>
> [c]    Counterfactual VQA: A cause-effect look at language bias. In CVPR.
>
> [d]    Overcoming Language Priors in Visual Question Answering with Adversarial Regularization. In NeurIPS.
>
> [e]    MUTANT: A training paradigm for out-of-distribution generalization in visual question answering. In EMNLP.
>
> [f]     Roses Are Red, Violets Are Blue... but Should Vqa Expect Them To? In CVPR.

---

> > ### Comment · Reviewer_pgR5 · 2021-08-25
> > **Thanks for the response**
> >
> > Thanks for writing a detailed response to my queries. Excited to see the VQA v2 and GQA-OOD results and ablations, and the conclusions based on that.
> > Though I am somewhat concerned about 78.x% number for LXMERT+D-VQA. Can you make sure that the LXMERT backbone is not using validation set itself in the training data or if validation set is not exposed in some form during training. The number is very high compared to normal accuracies for LXMERT. Ideally, can you instead evaluate on test-dev for the final version? (If possible, can you respond to me with test-dev numbers?). Furthermore, since you mentioned that LXMERT is from original repo, it is probably trained on val set as well. Please make sure to fix this.
> >
> > Most of my concerns are resolved and I am increasing my rating to 6 under the impression that these changes will be added to the camera ready paper if accepted.

---

> > > ### Author Response · Authors · 2021-08-30
> > > **Response**
> > >
> > > Thanks for your reminder. We carefully re-check up the official GitHub repository of the LXMERT, and find that the pre-training process of the LXMERT would use part of the validation set. However, the result of 78.8% is obtained by evaluating our D-VQA + LXMERT on all of the validation set, and thus our D-VQA + LXMERT has higher performance. To alleviate this issue, we adopt the minival split to further evaluate our method, where the minival split is not used in the pre-training process and is obtained from the official GitHub repository of the LXMERT. To further alleviate the concerns of the reviewers, we also conduct experiments on the test split of the VQA v2 dataset. All the results are shown below. We also release our results on the [leaderboard of VQA Challenge 2021](https://eval.ai/web/challenges/challenge-page/830/leaderboard/2278).
> > >
> > > ---
> > > Results on the minival split of the VQA v2 dataset
> > >
> > > |Model| All |
> > > |:---:|:----:|
> > > |LXMERT| 70.2 |
> > > |LXMERT + D-VQA(Ours)| 69.8 |
> > >
> > > ---
> > > Results on the Test-Standard split of the VQA v2 dataset
> > >
> > > |Model| All | Yes/No | Num | Other|
> > > |:---:|:----:|:----:|:----:|:----:|
> > > |LXMERT| 72.54 | 87.97 | 54.94 | 63.13 |
> > > |LXMERT + D-VQA(Ours)| 72.17 | 88.29 | 52.48 | 62.65 |
> > >
> > > ---
> > > Results on the Test-Dev split of the VQA v2 dataset
> > >
> > > |Model| All | Yes/No | Num | Other|
> > > |:---:|:----:|:----:|:----:|:----:|
> > > |LXMERT| 72.42 | - | - | - |
> > > |LXMERT + D-VQA(Ours)| 72.11 | 88.23 | 52.97 | 62.7 |
> > >
> > > ---
> > >
> > > From these results, our LXMERT+D-VQA achieves comparable performance with the LXMERT, which demonstrates our D-VQA is able to alleviate the bias issue, while maintaining the performance on the in-domain dataset.

---

### Decision · Program_Chairs · 2021-09-27

**Decision:**

Accept (Poster)

**Comment:**

The paper proposed a new method for debiasing VQA models. A consensus for acceptance emerged very quickly in the discussion phase, helped by a swift rebuttal by the authors and additional experiments on new datasets.

The AC concurs.